# From displacement to hunger: How migration due to conflict affects food security in Yemen

**Sayed Jubair Bin Hossain**[1], **Niaz Makhdum**[2], **Maruf Hasan Rumi**[3]*

**1** Department of Economics, University of Dhaka, Dhaka, Bangladesh, **2** Department of Public Administration, Begum Rokeya University, Rangpur, Bangladesh, **3** Department of Public Administration, University of Dhaka, Dhaka, Bangladesh

* marufhasanrumi@du.ac.bd

## Abstract

### Background

Conflict-induced migration is a key driver of global food insecurity. Over the past decade, prolonged conflicts have displaced millions of people. This crisis plunged millions into acute hunger, deepening existing vulnerabilities and severely restricting access to food and livelihoods across Yemen.

### Objective

This study aims to investigate the relationship between conflict-induced migration and food insecurity in Yemen, focusing on how displacement exacerbates hunger and malnutrition among affected populations.

### Methodology

This study analyzed responses from 13,891 individuals across 22 governorates, using data from the Yemen Multiple Indicator Cluster Survey (MICS) 2022–2023. The two primary indicators of food insecurity — specifically, hunger frequency and instances of not eating for a full day — were examined. Additionally, Negative Binomial Regression was employed to model the over-dispersed count data while controlling for factors such as sex, age, education, household size, urban-rural location, and wealth.

### Results

The findings revealed that approximately 24% of respondents experienced hunger, while about 12% went without food for an entire day. Moreover, conflict-induced migration substantially augmented the incidence of hunger (IRR = 1.262; 95% CI: 1.111–1.435) and going without eating for an entire day (IRR = 1.251; 95% CI: 1.032–1.516), though international migration to Yemen decreased food insecurity.

**Data availability statement:** https://ghdx. healthdata.org/record/yemen-multiple-indica- tor-cluster-survey-2022-2023.

**Funding:** The author(s) received no specific funding for this work.

**Competing interests:** The authors have declared that no competing interests exist.

Additionally, being female, having higher education, and belonging to wealthier households were protective factors against food insecurity, while urban residence significantly increased the likelihood of hunger (IRR = 2.231; 95% CI: 1.990–2.501) and meal skipping (IRR = 3.657; 95% CI: 3.086–4.335).

## Conclusion

Conflict-induced displacement is associated with worsened food insecurity, escalating hunger, and deprivation among displaced people, underscoring the immediate necessity of targeted humanitarian interventions in Yemen. Policy measures must prioritize the re-establishment of food systems and livelihoods in affected regions to alleviate the intensifying hunger crisis.

## Introduction

Migration caused by conflict has become a leading global crisis of the 21st century, with severe socioeconomic and humanitarian effects. The armed conflict in Yemen, ongoing since 2014, has resulted in the displacement of millions of people, both internally and externally [1]. The total number of people living in internal displacement increased by 51% over the past five years, reaching a record high of 75.9 million people across 116 countries at the end of 2023. Of the 75.9 million Internally Displaced People (IDPs) at the end of 2023 globally, 68.3 million were displaced by conflict and violence, and in Yemen, this number is 4.5 million [2]. Forced displacements can create short- to medium-term impacts on food security in local communities in mainly two ways [3]. First, IDPs can cause a population shock, a sudden increase in population due to forced displacement in a geographical area [4]. Such a population shock can also create additional pressure on existing food systems. Hence, the increasing number of IDPs in Yemen has also escalated the pre-existing vulnerabilities, especially in terms of food insecurity. As a country that has always been one of the poorest in the Arab world, Yemen has, for a long time, had to struggle with poverty, unemployment, and inadequate food systems [5]. However, the introduction of the conflict made the predicament unbearable by causing obstructed access to food. The plight of people, particularly those who are affected by the conflict, has leveraged these problems, and as a consequence, hunger and malnutrition have become a significant threat to the displaced individuals and host communities [5,6]. The hunger problem in Yemen is extremely severe. Based on the United Nations (2022) report, approximately 17 million people, or more than half of Yemen's population, have experienced food insecurity.

In contrast, the majority of them were displaced due to the conflict, and the rest have limited availability of healthy food, thereby becoming more vulnerable to chronic hunger and severe malnutrition [7]. The situation is often made worse by migration forced on those displaced people, who not only lose the income they had previously had as farmers and in small businesses but also the ability to become self-reliant. As a consequence, the movement of people (either within Yemen or to the neighboring

countries) directly affects the supply of food, jobs, and social benefits, which are essential elements in maintaining the food security status of the people [8].

The dynamics of food insecurity and migration are intricately intertwined and challenging to disentangle. Conflict-driven migration within Yemen and the international movement to the neighboring states have diverse food security challenges [6,9]. Migrants not only lose connection to the agricultural sector, animals, and markets but also face income and social barriers when they reach the host areas [10]. It has been demonstrated in earlier studies that the problem of food insecurity is more prevalent among conflict migrants than among non-migrants or voluntary migrants, mainly due to the loss of social and economic capital, as well as disruptions to food systems [11]. The evidence from other regions affected by conflict, such as Syria, Nigeria, and South Sudan, indicates that forced migration has led to a significant reduction in food security, with displaced people experiencing more intense hunger than the native communities [12,13]. Similarly, in Yemen, the displacement caused by the conflict has been a major contributor to the increase in food insecurity [14].

Nonetheless, this critically urgent issue has not even been addressed in the academic realm. The purpose of this study is to investigate the associations between conflict-induced migration and food insecurity in Yemen, using data from the Yemen Multiple Indicator Cluster Survey (MICS) 2022–2023 [15]. The paper aimed to establish a connection between migration resulting directly from a crisis and food insecurity by examining and quantifying two key indicators: Hunger rates and the frequency of daily meal shortages, particularly in conflict-affected areas. Meanwhile, the study would also control socio-demographic factors, including gender, education, and urban-rural location, which may play a role in this relationship. The study's insights would create effective measures for addressing hunger and malnutrition in the refugee population and would be an influential factor in humanitarian response plans in Yemen and other conflict-affected areas worldwide.

## Methodology

### Data description

The data used in this study were from the Yemen Multiple Indicator Cluster Survey 2022–2023. Yemen's Central Statistical Organization led this comprehensive survey with technical assistance from UNICEF. The survey employed a multi-stage, stratified cluster sampling design. The survey was designed to gather an extensive set of data to assess the living conditions of Yemenis. The survey was conducted at sub-national and rural/urban levels for the twenty-two governorates in Yemen. Due to ongoing conflict and access limitations, minor modifications in some areas were incorporated into the sampling plan. Approximately 20,000 households were targeted for the survey, and the survey questionnaire collected data on migration status and food insecurity. The migration and food security module was administered to 13,891 individuals. The total number of observations analyzed in this study is 13,891 individuals. The data was available upon registration from MICS survey website.

### Dependent and independent variables

The two principal dependent variables investigated in this study are the incidence of hunger and frequency of not eating for a whole day. The survey questionnaire inquired whether the respondent had been hungry but did not eat due to a lack of money or other resources in the past month and whether the respondent had gone without eating for a whole day because of lack of money or other resources during the same period. These two variables were specified as categorical variables. Both variables were presented as incidences of hunger or not eating for an entire day in the past 30 days (the indicators are count variables). The hunger indicator was coded based on the frequency with which an individual experiences hunger each week: Never (coded as 0), once a week (coded as 1), more than once a week (coded as 2), and almost every week (Coded as 3). For this study, these categories were grouped into No hunger (coded as 0), mild hunger (coded as 1), moderate hunger (coded as 2), and severe hunger (Coded as 3). The Not Eating for a Whole Day indicator was similarly coded. For this study, these categories were coded as follows: Never (coded as 0), Rarely (coded as 1), Sometimes (coded as 2), and Often (Coded as 3).

The focus of this study was how migration due to conflict affects the incidence of food insecurity among Yemeni individuals. The study employed five binary migration variables to examine the relation between migration and food insecurity. The variables are listed below. The MICS survey questionnaire included a module on the respondent's migration status. Two questions were the focus of the study. First, the respondents were asked where they migrated from. The second question was the reason for migration. Using these questions, five variables were created to analyze the effect of migration on food security.

1. Migration due to conflict: This binary variable equals '1' if the individual migrated due to conflict (as self-reported, regardless of whether the conflict was internal or international) and '0' otherwise.

2. Migrating from another country: This binary variable equals '1' if the individual migrated from outside Yemen (as self-reported, regardless of the reason) and '0' otherwise.

3. Migrating from another Governorate: This binary variable equals '1' if the individual migrated from within Yemen (self-reported, regardless of the reason) and '0' otherwise.

4. Migrating due to conflict from Another Country: This binary variable equals '1' if the individual migrated due to conflict from another country to Yemen, and '0' otherwise.

5. Migrating due to conflict from another Governorate: This binary variable equals '1' if the individual migrated due to conflict from another Governorate in Yemen and '0' otherwise.

Additional variables were included as covariate. All the information is given in Table 1. These control variables include:

1. Female: binary variable; if the individual is a female, it's 1, and 0 if male.

2. Age: A categorical variable representing the age of the individual, ranging from under 10 years old to over 60 years old.

3. Household members: A categorical variable representing the number of members in the individual's household.

4. Education Level: A categorical variable representing the level of education the individual completed.

5. Urban: A binary variable where the individual lives in an urban area, it is 1 and 0 indicate the individual lives in a rural area.

6. Composite wealth index: A standardized continuous variable that represents the wealth level of the individual's household using asset ownership.

These variables were used in the statistical analysis to determine the association between migration, particularly migration due to conflict, and the incidence of food insecurity.

## Statistical analysis

The statistical analysis employed two methods to investigate the association of conflict migration on the incidence of food insecurity. The first part used descriptive statistics analysis, and the second part utilized Negative Binomial Regression analysis.

In the descriptive statistics analysis part, the sample was first illustrated. This displays the part of the sample that was chosen and provides a snapshot of the data used in this study. Secondly, descriptive statistics and Pearson's chi-square tests were used to study the differences in food insecurity among conflict migrants across governorates and age groups in Yemen.

The two dependent variables ('Hunger' and 'Not Eating for a Whole Day') are coded as incidences of food insecurity in a month. As the variables were count data, Poisson or Negative Binomial Regression were the best methods for

**Table 1. Descriptive Statistics.**

|  | N | % |
|---|---|---|
| **Sex of Respondent** |  |  |
| Male | 6,509 | 46.86 |
| Female | 7,382 | 53.14 |
| **Area** |  |  |
| Urban | 8,151 | 58.68 |
| Rural | 5,740 | 41.32 |
| **Governorate** |  |  |
| Ibb | 1,377 | 6.57 |
| Abyan | 1,282 | 6.12 |
| Sana'a City | 4,087 | 19.51 |
| Al Bayda | 838 | 4.00 |
| Taizz | 1,094 | 5.22 |
| Al Jawf | 302 | 1.44 |
| Hajjah | 490 | 2.34 |
| Al Hudaydah | 536 | 2.56 |
| Hadramaut | 556 | 2.65 |
| Dhamar | 621 | 2.97 |
| Shabwah | 735 | 3.51 |
| Sa'ada | 1,861 | 8.89 |
| Sana'a | 540 | 2.58 |
| Aden | 1,913 | 9.13 |
| Lahj | 345 | 1.65 |
| Marib | 688 | 3.28 |
| Al Mahwit | 725 | 3.46 |
| Al Maharah | 312 | 1.49 |
| Amran | 989 | 4.72 |
| Al Dhale'e | 478 | 2.28 |
| Raymah | 955 | 4.56 |
| Socotra | 220 | 1.05 |
| **Age** |  |  |
| 0-10 Years | 1325 | 9.54% |
| 11 - 20 Years | 3760 | 27.07% |
| 21 - 30 Years | 3803 | 27.38% |
| 31-40 Years | 2658 | 19.13% |
| 41-50 Years | 1436 | 10.34% |
| 51-60 Years | 606 | 4.36% |
| 61 and Above Years | 303 | 2.18% |
| **Education Level** |  |  |
| Pre-primary or none | 45 | 0.32 |
| Primary Education | 5,414 | 38.97 |
| Lower Secondary Education | 2,878 | 20.72 |
| Upper Secondary Education | 3,647 | 26.25 |
| Higher | 1,891 | 13.61 |
| **Wealth Quintile** |  |  |
| Poorest | 1,151 | 8.29 |
| Second | 1,204 | 8.67 |

*(Continued)*

**Table 1.** (Continued)

| | N | % |
|---|---|---|
| Middle | 1,959 | 14.1 |
| Fourth | 4,015 | 28.9 |
| Richest | 5,562 | 40.04 |
| **Hunger (In the past month)** | | |
| No Hunger | 10,536 | 75.85 |
| Mild Hunger | 1,293 | 9.31 |
| Moderate Hunger | 1,135 | 8.17 |
| Severe Hunger | 927 | 6.67 |
| **Without Eating for a whole day (In the past month)** | | |
| Never | 12,243 | 88.14 |
| Rarely | 784 | 5.64 |
| Sometimes | 536 | 3.86 |
| Often | 328 | 2.36 |

regression analysis. The dependent variables were over-dispersed, indicating that the variance of the indicators exceeds their mean. For this reason, a Negative Binomial Regression method is the most suitable approach in this regard. To investigate if the migration variables with statistical significance explain the model of food insecurity, Akaike Information Criterion, Bayesian Information Criterion, and Likelihood Ratio Test were used (e.g., see Supporting Information, Appendix Table 1). The results of the regression analysis are presented as the Incidence Rate Ratio (adjusted for differences in governorates), along with the corresponding p-value and 95% confidence interval.

## Results

Table 1 presents a snapshot of the sample's characteristics used for analysis. In the study, the distribution was almost equal between male (46.86%) and female (53.14%) respondents. There was a slightly greater number of respondents living in urban areas (58.68%) than in rural areas (41.32%). The highest sample of respondents resided in Sana's City (19.51%), Aden (9.13%), Sa'ada (8.89%), Ibb (6.57%), and Abyan (6.12%) governorates. About 54% of the respondents were aged between 11 and 13 years old. About 61% of the respondents had at least a Lower Secondary Level of Education. The highest proportion of the sample was in the wealthiest quintile (40.04%) of households. About 24% of the respondents experienced some form of hunger, and about 12% of the respondents, in some frequency, did not eat for a whole day.

Table 2 presents the level of hunger across the governorates. It shows that the highest levels of severe hunger among conflict migrants were in Al Mahwit (44.94%) and Dhamar (29.03%). The highest levels of mild hunger were in Al Jawf (47.62%), Al Hudaydah (34.44%), and Al Maharah governorates. The highest levels of no hunger among conflict migrants were from the Ibb (77.27%), Abyan (73.36%), and Taizz (68.24%) governorates.

Table 3 presents the frequency of going without eating for a whole day across governorates that have migrated due to conflict. Conflict migrants in Lahj (19.51%) and Al Jawf (14.29%) had the highest proportion of reported often going without eating for a whole day. The highest reported levels of rarely going without eating for an entire day were among conflict migrants in Al Jawf (47.62%), Al Maharah (30.61%), and Al Dhale'e (27.45%).

Table 4 shows that children under 18 experienced a higher level of hunger (36.20%) than other adult age groups who migrated due to conflict. Elderly individuals had the highest level of severe hunger (13.16%). Similarly, Table 5 also shows that conflict migrant children (22.47%) faced a higher frequency of not eating for a whole day compared to other age groups. Elders faced the most often of not eating for a whole day compared to other groups (5.26%).

**Table 2. Level of Hunger across governorates that migrated due to conflict.**

|  | No Hunger | Mild Hunger | Moderate Hunger | Severe Hunger |
|---|---|---|---|---|
| Al Mahwit | 39.33% | 8.99% | 6.74% | 44.94% |
| Dhamar | 46.77% | 11.29% | 12.90% | 29.03% |
| Al Jawf | 0.00% | 47.62% | 23.81% | 28.57% |
| Al Bayda | 51.69% | 11.02% | 9.32% | 27.97% |
| Marib | 50.46% | 25.23% | 4.62% | 19.69% |
| Al Dhale'e | 58.82% | 13.73% | 7.84% | 19.61% |
| Ibb | 77.27% | 1.82% | 3.64% | 17.27% |
| Lahj | 63.41% | 7.32% | 12.20% | 17.07% |
| Aden | 53.02% | 8.05% | 22.15% | 16.78% |
| Amran | 38.53% | 14.68% | 32.11% | 14.68% |
| Sana'a | 61.22% | 4.08% | 20.41% | 14.29% |
| Hajjah | 61.17% | 6.80% | 22.33% | 9.71% |
| Raymah | 57.41% | 18.52% | 15.74% | 8.33% |
| Taizz | 68.24% | 16.47% | 7.65% | 7.65% |
| Al Hudaydah | 38.89% | 34.44% | 21.11% | 5.56% |
| Abyan | 73.36% | 9.17% | 12.01% | 5.46% |
| Al Maharah | 57.14% | 30.61% | 8.16% | 4.08% |
| Sana'a City | 91.24% | 1.38% | 4.61% | 2.76% |
| Shabwah | 91.18% | 0.00% | 7.35% | 1.47% |
| Sa'ada | 80.16% | 14.49% | 4.25% | 1.10% |
| Hadramaut | 91.30% | 0.00% | 8.70% | 0.00% |
| Socotra | 100.00% | 0.00% | 0.00% | 0.00% |
| Pearson Chi2 p-value | < 0.001 |  |  |  |

Table 6 presents the results of the negative binomial regression, illustrating the link between experiencing hunger or going without food for a whole day and Yemeni individuals. Individuals who migrated due to conflict were significantly and positively associated with a higher incidence of hunger (IRR: 1.262, 95% CI: 1.111–1.435) and reported not eating for the entire day (IRR: 1.251, 95% CI: 1.103–1.516). However, individuals migrating to Yemen had a lower incidence of food insecurity (Hunger: IRR: 0.638, 95% CI: 0.480–0.848; Without Eating: IRR: 0.600, 95% CI: 0.367–0.981). Migrating from another governorate did not have a statistically significant association with the incidence of food insecurity. Individuals who migrated due to conflict and from another country, on average, had a higher correlation of incidence of hunger (IRR: 2.100, 95% CI: 1.279–3.450) by approximately more than two-fold and the possibility of going without eating (IRR: 6.001, 95% CI: 2.951–12.206) by about six-fold. Migrating due to conflict from another governorate did not have a statistically significant relation to the incidence of hunger among individuals. Migrating due to conflict from another governorate was associated with a statistically significant positive correlation on the incidence of an individual going without eating for a whole day (IRR: 1.312, p-value < 0.050).

Among the confounding variables, being female is associated with a lower incidence of going hungry (IRR: 0.871, 95% CI: 0.812–0.935) or not eating for a whole day (IRR: 0.782, 95% CI: 0.700–0.873). The education level had a statistically significant correlation on decreasing the incidence of hunger (IRR: 0.897, 95% CI: 0.867–0.929) and going without eating for a whole day (IRR: 0.867, 95% CI: 0.821–0.916). Similarly, if the individual is from a wealthier household, there is significantly correlated with lower incidence of hunger (IRR: 0.526, 95% CI: 0.491–0.564) and going without eating for a whole day (IRR: 0.436, 95% CI: 0.396–0.481). Individuals living in urban areas were, on average, 2.231- and 3.657-times

**Table 3. Frequency of going without eating for a whole day across governorates that migrated due to conflict.**

| | Never | Rarely | Sometimes | Often |
|---|---|---|---|---|
| Lahj | 68.29% | 4.88% | 7.32% | 19.51% |
| Al Jawf | 0.00% | 47.62% | 38.10% | 14.29% |
| Amran | 53.21% | 24.77% | 9.17% | 12.84% |
| Marib | 70.15% | 14.46% | 3.08% | 12.31% |
| Aden | 79.87% | 1.34% | 7.38% | 11.41% |
| Sana'a | 73.47% | 2.04% | 14.29% | 10.20% |
| Taizz | 81.76% | 7.65% | 2.35% | 8.24% |
| Abyan | 84.50% | 2.18% | 8.95% | 4.37% |
| Al Bayda | 81.36% | 3.39% | 11.02% | 4.24% |
| Ibb | 94.55% | 0.00% | 1.82% | 3.64% |
| Al Hudaydah | 82.22% | 12.22% | 2.22% | 3.33% |
| Dhamar | 75.81% | 19.35% | 3.23% | 1.61% |
| Shabwah | 94.12% | 0.00% | 4.41% | 1.47% |
| Sa'ada | 88.03% | 9.45% | 1.42% | 1.10% |
| Al Mahwit | 59.55% | 12.36% | 28.09% | 0.00% |
| Raymah | 73.15% | 12.04% | 14.81% | 0.00% |
| Hajjah | 73.79% | 11.65% | 14.56% | 0.00% |
| Al Dhale'e | 58.82% | 27.45% | 13.73% | 0.00% |
| Hadramaut | 91.30% | 0.00% | 8.70% | 0.00% |
| Al Maharah | 61.22% | 30.61% | 8.16% | 0.00% |
| Sana'a City | 97.70% | 0.00% | 2.30% | 0.00% |
| Socotra | 100.00% | 0.00% | 0.00% | 0.00% |
| Pearson Chi2 p-value | < 0.001 | | | |

**Table 4. Level of Hunger across age groups that migrated due to conflict.**

| | Age and Hunger level among Conflict migrants | | | |
|---|---|---|---|---|
| | No Hunger | Mild Hunger | Moderate Hunger | Severe Hunger |
| Children (18 and below) | 63.80% | 14.35% | 10.89% | 10.97% |
| Young adult [19–30] | 68.60% | 12.91% | 7.91% | 10.58% |
| Adult [31–64] | 67.33% | 11.62% | 11.32% | 9.72% |
| Elderly (65+) | 71.05% | 7.89% | 7.89% | 13.16% |
| Pearson Chi2 p-value | < 0.001 | | | |

higher incidence to go hungry (IRR: 2.231, 95% CI: 1.990–2.501) or go without eating for a whole day (IRR: 3.657, 95% CI: 3.086–4.335), respectively.

## Discussion

Conflict-based displacement can worsen food insecurity, but this association has not been examined among people in Yemen specifically. Keeping this in mind this study was designed to examine the association of war and conflict-induced migration on food security in Yemen, utilizing data from the Yemen Multiple Indicator Cluster Survey (MICS) 2022–2023.

The investigation revealed that migration, particularly for conflict-related reasons, had a significant correlation on both the prevalence of hunger and the frequency of going without food for an entire day. The results align with the findings

**Table 5. Frequency of going without eating for a whole day across age groups that migrated due to conflict.**

|  | Age and Frequency of not eating for a whole day among Conflict migrants | | | |
|  | Never | Rarely | Sometimes | Often |
|---|---|---|---|---|
| Children (18 and below) | 77.55% | 9.45% | 8.02% | 4.98% |
| Young adult (19 to 30) | 80.93% | 8.26% | 6.86% | 3.95% |
| Adult (31 to 64) | 82.87% | 7.82% | 4.61% | 4.71% |
| Elderly (65+) | 84.21% | 7.89% | 2.63% | 5.26% |
| Pearson Chi2 p-value | <0.001 | | | |

**Table 6. The result of the Negative Binomial Regression.**

|  | Hunger | | | Without Eating | | |
|  | IRR | 95% Confidence Interval | P-value | IRR | 95% Confidence Interval | P-value |
|---|---|---|---|---|---|---|
| Migration due to conflict | 1.262 | (1.111, 1.435) | < 0.001 | 1.251 | (1.032, 1.516) | < 0.050 |
| Migrating from another country | 0.638 | (.480,.848) | <0.010 | 0.600 | (.367,.981) | < 0.050 |
| Migrating from another Governorate | 1.003 | (.914, 1.099) | NS | 0.889 | (.761, 1.039) | NS |
| Migrating due to conflict from another Country | 2.100 | (1.279, 3.450) | <0.001 | 6.001 | (2.951, 12.206) | < 0.001 |
| Migrating due to conflict from another Governorate | 1.014 | (.864, 1.188) | NS | 1.312 | (1.032, 1.667) | < 0.050 |
| **Control variables** | | | | | | |
| Female | 0.871 | (.812,.935) | < 0.001 | 0.782 | (.700,.873) | < 0.001 |
| Age | 0.998 | (.996, 1.001) | NS | 0.995 | (.990,.999) | < 0.010 |
| Household Members | 1.002 | (.992, 1.012) | NS | 0.996 | (.981, 1.011) | NS |
| Education Level | 0.897 | (.867,.929) | < 0.001 | 0.867 | (.821,.916) | < 0.001 |
| Urban | 2.231 | (1.990, 2.501) | < 0.001 | 3.657 | (3.086, 4.335) | < 0.001 |
| Composite Wealth Index | 0.526 | (.491,.564) | < 0.001 | 0.436 | (.396,.481) | < 0.001 |

IRR: Incidence Rate Ratio (Adjusted for Governorate differences).

of the current literature, which primarily emphasize the detrimental effects of forced migration on food security. It is well-documented that the exodus of populations from their homes and their relocation significantly increases their likelihood of experiencing food insecurity, as they are deprived of access to the means of earning their living, food, and other social services [11].

The findings revealed that people displaced due to conflict experienced, on average, a significantly higher incidence of hunger and skipping meals throughout the day. This provides support to the hypothesis that conflict-led displacement reduces food accessibility, deepens economic instability, and intensifies the need for humanitarian aid, which is often insufficient in the majority of cases [12]. The results align with research from other conflict-affected areas, such as Syria and South Sudan, where involuntary migration has been found to contribute to higher food insecurity due to the collapse of local economies, mass displacement from agricultural land, and the loss of income-generating activities [13].

The data also indicate that the situation of people moving from one country to another due to war or conflict is the most serious problem in food security, with a significantly higher rate of starvation and people not having access to food for a day. This result is consistent with previous studies that showed that international migrants are usually plagued with extra social and economic woes like legal obstacles, discrimination, and exclusion from the social safety net, which underpin their food and nutrition insecurity [8]. These issues may also emerge as a result of the struggles migrants face when integrating into communities that are not easily accessible to the host communities, leading to an aggravation of their food insecurity [11].

However, it was found that migration from another part of the governorate within Yemen did not have a statistically significant effect on the incidence of lack of food or going without food for the entire day, which suggests that local displacement within a country, on average, may not have a relationship as severe as international migration on the food security of a nation. On the contrary, migration due to conflict from a different governorate was statistically significant relationship for the incidence of going without a meal for an entire day, thus showing that even internal displacement can affect access to food, albeit with a lesser correlation compared to international migration. These results are consistent with previous literature on food insecurity and migration. Migration driven by conflicts often results in the loss of access to food, which alters social and economic structures and places additional burdens on host communities [11]. There is also evidence that these incidences of food insecurity are exacerbated in vulnerable regions, such as Africa, the Middle East, and Eastern Europe [11]. Research on forced migrants in Nigeria and Jordan shows that migrants lose their social, cultural, human, and economic capital, which is essential to maintaining food security. It has also been demonstrated that the influx of migrants into host communities strains both cultural and economic capital within these communities [12,13].

Among the determinants, a few variables were observed to be confounding factors of food insecurity as well. It was noticeable that the female participants were statistically less prone to going to sleep hungry or starving for a whole day. This suggests that women may be the primary figures in eliminating food insecurity in families, as they are typically responsible for household food management [16–18]. This result is similar to many other studies that have recognized women as the main authority in handling the food security of their households, especially in areas with constrained resources [19].

The influence of education on food security was also observed, as respondents with higher levels of education had a lower risk of hunger and food insecurity. This finding aligns with earlier research that has emphasized the role of education in promoting economic stability, access to resources, and nutritional knowledge, all of which are key factors in preventing food insecurity [20–23]. Educational attainment reduces food insecurity by equipping individuals with literacy skills, nutrition education, and increasing their likelihood of financial stability [19]. On the same note, a statistically significant decrease in the ratio of food insecurity was observed in individuals with higher incomes, thus implying the widely observed connection between income and food security [12].

Living in the urban areas led to a higher incidence of hunger and going without food as urban migrants face numerous hurdles in these cities, where the scarcity of resources is severe, and the number of self-sufficiency opportunities in food production is low [24,25]. A multitude of studies have indicated that employment issues are the primary cause of stress for urban migrants, and due to the escalating living costs associated with the job, they struggle more to access sufficient food [26–28]. The rural-urban division of food insecurity highlights the importance of understanding the role of the social environment in the interrelationship between migration and food insecurity. Previous literature has portrayed urban sprawl and inefficient city structures in sub-Saharan Africa as leading to socioeconomic segregation, which affects food accessibility for migrants [29].

Finally, the research explored that the wealth index not only had a significant protective effect on food security but also those citizens who were better off in terms of social class on average were the ones who encountered fewer physical sensations of hunger and no experiences of missing a meal. This highlights the importance of financial well-being in achieving food security, especially in conflict-affected areas where access to food is often highly constrained [30].

The research has highlighted that displacements caused by war and conflict have a strong positive relationship with food insecurity. The findings of this study have significant policy and humanitarian implications, indicating the necessity of direct aid for conflict migrants and those migrating from abroad, in particular, and suggesting that addressing the deeper socioeconomic issues that lead to food insecurity in Yemen is also necessary.

## Conclusion and policy recommendation

The movement of people due to war and conflict presents a challenge not only for the internal inhabitants but also for those crossing borders, which in turn seems to influence the severity of the problem in the latter case. The findings of this

study also indicate that factors such as sex, education, wealth, and the urban-rural location of the population are highly significant in determining food security. According to the findings of this study, women, individuals with higher education, and the affluent were not the minority of people who faced food scarcity; however, urban dwellers showed an increasing vulnerability. This evidence is consistent with the existing literature, which indicates that conflicts can hinder food access, leading to economic instability and necessitating more humanitarian assistance. This research highlights the complex issue of food insecurity in conflict-prone territories, underscoring that migration, particularly forced migration, is a significant contributor to people's exposure to risk of food insecurity. In Yemen, food access interruptions due to displacement are accompanied by social and economic disparities that, in turn, exacerbate the difficulties the displaced face. It is for these reasons that addressing the issue of hunger among migrants should be a comprehensive process to provide both prompt relief and sustainable solutions simultaneously. Future research should investigate the pathways through which conflict-induced migration affects food insecurity.

Several policy recommendations flow from this research to mitigate the food insecurity faced by the conflict migrants in Yemen. The first and most pressing issue is that humanitarian programs should be boosted, with a focus on the migrant population, especially foreign migrants. These groups face numerous unprecedented challenges that hinder their access to food, stemming from legal, social, and economic inequalities. Thus, the aid program should not only deliver essential food items in an emergency but also explore long-term food security strategies, such as providing nutritional treatments and establishing a sustainable food distribution system in the region.

Second, it is essential to focus on creating sustainable jobs for the people forced to leave, which is particularly crucial in cities where migrants are at the highest risk. In the pursuit of a sustainable solution to Yemen's food crisis, significant efforts can be made under the nation's National Food Security Strategy. This involves securing large-scale investment in urban centers from both governmental and foreign sources. This investment should be strategically directed towards initiatives that will create jobs and diversify and expand markets, ultimately fostering a more resilient and self-sufficient food system.

Thirdly, enhancing the availability of education, particularly for women and children, has paramount importance for sustainable food security. Knowledge gained from education enables people to secure decent and stable incomes and access to better nutrition and food security. Hence, educational interventions should incorporate elements of both formal education and training in nutrition, agricultural practices, and food security, thereby enabling displaced individuals to possess the necessary information for informed decision-making regarding provision of food. Moreover, assimilating inclusive strategies into food security and migration policies is crucial. Women, children, and elderly people have always been the major sufferers within families; therefore, if women are empowered through the provision of resources, education, and economic opportunities, they can substantially contribute to food security within their families, resulting in empowerment through access to land, credit, and entrepreneurial training. But this will need through consultation with religious and tribal leaders as patriarchal norms and rigid social structures will impede this attempt in Yemeni society.

In conclusion, this study reiterates the necessity for comprehensive approaches that address the pressing issue of food security for displaced people in Yemen in a sustainable manner. Humanitarian aid, job creation, skill training, and expansion of economy through massive investment are to be combined into a comprehensive strategy for food security intervention. The implementation of these recommendations can positively influence the resilience of the affected communities, the reduction of food insecurity, and the recovery of Yemen and similar conflict-affected regions in the long run by the governments and humanitarian organizations.

## Supporting information

**Appendix Table 1. Post-estimation tests.**
(DOCX)

## Author contributions

**Conceptualization:** Sayed Jubair Bin Hossain, Niaz Makhdum, Maruf Hasan Rumi.

**Formal analysis:** Sayed Jubair Bin Hossain, Maruf Hasan Rumi.

**Investigation:** Sayed Jubair Bin Hossain, Maruf Hasan Rumi.

**Methodology:** Sayed Jubair Bin Hossain, Niaz Makhdum, Maruf Hasan Rumi.

**Software:** Sayed Jubair Bin Hossain.

**Validation:** Sayed Jubair Bin Hossain, Niaz Makhdum.

**Visualization:** Sayed Jubair Bin Hossain.

**Writing – original draft:** Sayed Jubair Bin Hossain, Niaz Makhdum, Maruf Hasan Rumi.

**Writing – review & editing:** Sayed Jubair Bin Hossain, Niaz Makhdum, Maruf Hasan Rumi.

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
