## [Decision Letter · Decision Letter 0]

12 Jun 2025

Dear Dr. Rumi,

We look forward to receiving your revised manuscript.

Kind regards,

António Raposo

Academic Editor

PLOS ONE

2. We notice that your supplementary table (Appendix Table 1) is included in the manuscript file. Please remove them and upload them with the file type 'Supporting Information'. Please ensure that each Supporting Information file has a legend listed in the manuscript after the references list.

Reviewers' comments:

Reviewer's Responses to Questions

**Comments to the Author**

1. Is the manuscript technically sound, and do the data support the conclusions?

Reviewer #1: Yes

Reviewer #2: Partly

Reviewer #3: Partly

Reviewer #4: Yes

2. Has the statistical analysis been performed appropriately and rigorously?

Reviewer #1: Yes

Reviewer #2: Yes

Reviewer #3: Yes

Reviewer #4: Yes

3. Have the authors made all data underlying the findings in their manuscript fully available?

Reviewer #1: Yes

Reviewer #2: Yes

Reviewer #3: Yes

Reviewer #4: Yes

4. Is the manuscript presented in an intelligible fashion and written in standard English?

Reviewer #1: Yes

Reviewer #2: No

Reviewer #3: Yes

Reviewer #4: Yes

Reviewer #1: The study provides robust evidence on the impact of conflict-induced migration on food security in Yemen, with clear policy implications. While limitations exist (e.g., cross-sectional data precluding causal inference on long-term effects, potential underreporting in high-conflict areas), these are common in conflict-region research and do not undermine the core conclusions.

Recommendation: Minor revisions are sufficient for acceptance. Specific suggestions include:

Expanding the discussion of limitations, such as data collection challenges in unstable regions and self-reporting biases .

Enhancing policy recommendations with granularity, e.g., targeting urban migrants and gender-sensitive interventions, given urban residence’s strong association with food insecurity (IRR = 2.231) and female protective effects (IRR = 0.871) .

Providing sub-group analyses (e.g., migration duration, age cohorts) in supplementary materials to deepen insights into heterogeneity.

The manuscript’s methodological rigor, data transparency, and timely relevance to humanitarian crises justify its publication in PLOS ONE following minor refinements.

Reviewer #2: • Introduction: The study objectives is mostly written in future tense.

• Ambiguity on unit of analysis: the authors described that 20,000 households were surveyed, but only 13,891 individuals were analyzed.

• The paper is about survey collection data on migration status and food insecurity," but there is no Module/questionnaire names and Specific items or questions used

• Grouped into No hunger, mild hunger. No clear explanation how frequency ranges were mapped into these levels.

• Migrating variables 1, 4, and 5 is overlapping. how conflict migration is distinguished across internal and international lines.

• Authors mentioned that migration is due to conflict is coded as “1 if yes,” but not how this was measured (survey question, self-report, time window).

• age education and household size are categorical variables but there is no number of categories mentioned similarly for continuous variables is it standardized?

• Sample illustration should be made clear

• There is inconsistency in P value like 0.050 is used, other times it is written < 0.001 or 0.010

• IRRs are 2.231 and 3.657, but interpreted too casually as "2 and 4 times

Reviewer #3: The manuscript reports the findings of an analysis of a large country-wide survey conducted among 22 regions in Yemen regarding population characteristics, migration, and food insecurity. The topic of food insecurity as related to conflict-induced migration is very important and has not been addressed for Yemen, which has experienced substantial conflict since 2014. The findings reveal a strong association between conflict-induced migration and greater incidence of experiencing hunger and days without eating, metrics of food insecurity. This study will make a good and needed contribution to the literature; however there are some major concerns that should be addressed, as detailed below.

1) A major limitation to the current presentation of the manuscript is that causal conclusions are drawn from a cross-sectional survey. Scientifically, it is not possible to draw causal conclusions from surveys that assess a single period of time. I do not doubt the many ways in which conflict-induced migration may impede access to regular meals and food needed to achieve/maintain food security; my concern is solely regarding the cross-sectional nature of the data. I believe this can be easily addressed in the manuscript; causal statements in the introduction, results, and conclusions can be amended to report on associations between conflict-induced migration and food insecurity.

2) One of the measures used to assess food insecurity in the study assessed 'hunger;' however hunger was not defined, nor were the instructions to respondents and question/response options defined. Based on the context of the study, it sounded like the authors sought to measure hunger over an extended period of time (that was not satisfied by being able to eat). However clearly specifying these details is necessary to understand and interpret the results.

3) Interpreting the effect size estimates, and moving beyond statements regarding significance/not, would further enhance the manuscript. For example, the findings that migration due to conflict from another country increased risk of hunger more than two-fold (IRR= 2.10) and increased risk of going without food for a day sixfold (IRR=6.001) is staggering and really conveys the extent of the risk to food security when individuals migrate due to conflict from another country. However, as currently presented the magnitude of the risk is not conveyed. I also think it would help to differentiate (or maybe even list first in the tables) migration due to conflict from another country vs migration due to conflict; the latter had effect size estimates that were far lower (~1.25), which is likely because that variable addresses migration due to conflict more broadly. I actually looked at the effect sizes first and was surprised they were not higher; I had to look at the table more closely to realize the variable re: migration due to conflict from another country was really the most prominent one in this case.

4) It appears that the authors responded "NA" to the question regarding whether this study was approved by an institutional review board. This caught my attention because the authors report in the descriptive statistics the 22 regions surveyed and their corresponding n's for respondents, some of which are quite small (n=~220). I have concerns that geographic location and responses regarding migration due to conflict could serve to risk identifying some individuals who may have completed the survey, especially if there are few respondents for a region and relatively small numbers of migrants due to conflict from other countries (eg, if ~220 participants completed the survey for a region and there are only ~200 migrants displaced due to conflict from another country).

5) Gender is reported as the label; however sex (male, female) are reported as the response options. Please clarify whether reporting is for gender identify or sex.

6) Some of the interpretation regarding females and food insecurity, and recommendations for policy solutions, address in a rather factual manner that females are 'the' ones responsible for food management and food security in the household. This is a pretty broad sweeping statement and more context should be provided to indicate to what degree this is the case for Yemen, as well as the surrounding region (given that the study addresses migration from other countries), and provide citations to support these statements, or amend them accordingly.

Reviewer #4: The presented study addresses a pertinent and actual concern of our time.

Some grammatical improvements would improve the text:

Imprecise timeframe: "From the last decade" is awkward. Prefer: “Over the past decade”; Repetition: The phrase “pushed them into acute hunger, exacerbating vulnerabilities and limiting access to food and livelihoods” is redundant and verbose. Streamlining would enhance clarity.

Scope narrowing needed: It begins globally but quickly shifts to Yemen. A smoother transition or immediate focus on Yemen would sharpen the narrative.

“Such as hunger frequency and…” — “such as” is inappropriate here. Use: “namely” or “specifically”.

“Investigate and show” is duplicative — use one.

The two indicators of food insecurity are simplistic. Explain how they were operationalized (e.g., from which MICS questions).

“Some level of hunger” is vague — this needs to be better defined.

Urban residence increasing food insecurity contradicts common assumptions. A brief explanation or hypothesis is warranted (e.g., displaced populations in urban informal settlements).

Precise confidence intervals (CIs) would make the statistics more informative.

The statement “general migration decreased food insecurity” needs unpacking. What constitutes “general migration”? Was it internal or international?

The phrase “worsens food insecurity” implies causality, which is not supportable given the cross-sectional design. Better: “is associated with worsened food insecurity.”

**Do you want your identity to be public for this peer review?** For information about this choice, including consent withdrawal, please see our Privacy Policy

Reviewer #1: No

Reviewer #2: No

Reviewer #3: **Yes: ** Tera L Fazzino

Reviewer #4: **Yes: ** M. João Lima

---

## [Author Response · Author response to Decision Letter 1]

3 Jul 2025

I have revised the file accordingly to the reviewers and editor comment. If any further revision is needed, I am available to do so.

---

## [Decision Letter · Decision Letter 1]

15 Jul 2025

Dear Dr. Rumi,

Thank you for submitting your manuscript to PLOS ONE. After careful consideration, we feel that it has merit but does not fully meet PLOS ONE’s publication criteria as it currently stands. Therefore, we invite you to submit a revised version of the manuscript that addresses the points raised during the review process.

We look forward to receiving your revised manuscript.

Kind regards,

António Raposo

Academic Editor

PLOS ONE

Journal Requirements:

Reviewers' comments:

Reviewer's Responses to Questions

**Comments to the Author**

Reviewer #1: (No Response)

Reviewer #2: All comments have been addressed

Reviewer #3: (No Response)

Reviewer #4: All comments have been addressed

2. Is the manuscript technically sound, and do the data support the conclusions?

Reviewer #1: Yes

Reviewer #2: Yes

Reviewer #3: Partly

Reviewer #4: Yes

3. Has the statistical analysis been performed appropriately and rigorously?

Reviewer #1: Yes

Reviewer #2: Yes

Reviewer #3: Yes

Reviewer #4: Yes

4. Have the authors made all data underlying the findings in their manuscript fully available?

Reviewer #1: No

Reviewer #2: Yes

Reviewer #3: No

Reviewer #4: Yes

5. Is the manuscript presented in an intelligible fashion and written in standard English?

Reviewer #1: Yes

Reviewer #2: Yes

Reviewer #3: No

Reviewer #4: Yes

Reviewer #1: Strengths and Improvements:

Methodological Rigor:Sampling strategy (multi-stage stratified cluster design) and variable operationalization (MICS questions for hunger/not eating) are now clearly defined.

Migration variables (e.g., conflict-induced international vs. internal) are distinct and analytically sound.

Statistical Robustness:Full reporting of IRR with 95% CIs (e.g., 1.262 [1.111–1.435]) and standardized *p*-values.

Model fit validated via AIC/BIC/Likelihood Ratio Tests (Appendix).

Language and Clarity:

Critical revisions: "worsens" → "associated with" (addressing causality), "gender" → "sex," and tense consistency (e.g., "aims to investigate").

Urban food insecurity paradox well-explained (resource competition/high costs; Refs 24–25).

Ethical Compliance:IRB waiver justification for secondary data use is appropriately stated.

Critical Revisions Required:

DATA AVAILABILITY (MUST ADDRESS):Revert to the original public repository link (https://ghdx.healthdata.org/record/yemen-multiple-indicator-cluster-survey-2022-2023).

"Data available on request" violates PLOS policy.

Minor Suggestions:

Subgroup Analyses:Briefly highlight key age/cohort findings (e.g., child migrants’ vulnerability) in the main text.

Urban Context:Strengthen policy linkages (e.g., connect "urban job creation" to Yemen’s National Food Security Strategy).

Policy Nuance:Specify how "gender-sensitive interventions" (p. 345) align with Yemen’s cultural frameworks.

Overall Recommendation:

This timely study makes a valuable contribution to conflict-related food insecurity literature. Accept after restoring the public data link. No further methodological or statistical revisions are needed.

Reviewer #2: 1. The objectives presented in the introduction should be in past tense.

2. Yemen’s Central Statistical Organization led this comprehensive survey with technical assistance from UNICEF. “were the authors part of these organizations?”

3. Line 133….is the 1st variable the sum of 4th and 5th one?

4. Line 183Sentence needs to be rephrased.

5. Line 187 represents or presents?

6. Line 195 reported experience of hunger

7. 202, 204, 211 sentence needs revision to make those clear.

8. 239 Either this or thus is enough

Reviewer #3: The authors present a revised version of the manuscript, which is in some instances improved. However, the authors did not address some of my most substantive feedback. My detailed comments are below.

1. The largest concern that I mentioned in my previous review that was not adequately addressed in this revision is regarding the use of causal claims and statements from cross-sectional data. The authors indicated in their response to reviews that they addressed this point by removing causal language throughout. However, this is not the case and causal language is used to discuss the findings throughout. I have pointed out specific instances of such language, although the authors should keep in mind that this is not an exhaustive list:

Lines 159-161 in statistical analysis: describes use of "two methods to investigate the impact of conflict migration on incidence of food insecurity." Using the term impact indicates causal and time sequential analyses; this is not as it is a cross-sectional analysis. The statement follows that "descriptive statistical analysis and .... negative binomial regression" are the methods employed, neither of which can support causal conclusions in this capacity.

Lines 207-208 results: "Migrating from another governorate did not have a statistically significant impact on..."

Lines 214-215: "Migrating due to conflict from another governorate had a statistically significant impact on..."

Lines 232-233 discussion: "had a considerable impact on both the prevalence of hunger and the frequency of going without food for an entire day."

Lines 299-300: "The research has highlighted that displacements caused by war and conflict are the primary cause of the increased food insecurity in Yemen."

2. There are also multiple instances in which the writing would benefit from clarity and careful consideration of framing. For example, the entire first paragraph of the discussion (Lines 229-237) is written in a manner that it is assumed that displacement causes food insecurity; however the authors also present this as the premise they were testing in the study. If it is already a foregone conclusion, it seems like doing the study would be unnecessary. In this case, better specification of the general premise that conflict-based displacement can worsen food insecurity, but this association has not been examined among people in Yemen specifically, would better facilitate the discussion intro and framing.

3. In the results, some of what is presented in the tables is repeated in the text; please summarise in the text and leave details for tables.

4. Some statements in the results and throughout are awkward and difficult to understand. one example is lines 209-211: "If the individual migrated due to conflict and from another country, with statistical

robustness, it increases the risk of incidence of hunger (IRR: 2.100, 95% CI: 1.279–3.450) by

approximately more than two-folds...." what is statistical robustness in this context and why not instead just report the p value or CI?

5. Some of the methods and results is repetitive and should be streamlined. For example, Methodology section Lines 98-100 report the study purpose stated immediately above in the intro section.

6. Some of the language in the methods is imprecise and should be edited to improve clarity and specificity. For example, Line 143 "Additional variables are included to control mediating and moderating variables." There were no mediating or moderating analyses conducted. These variables were only included as covariates and therefore they should be labeled as such.

7. Throught the manuscript, the tense switches from reporting in present tense to past tense and back again. Reporting of methods and results, which were conducted in the past, should be reported in past tense.

8. In general, it is not necessary to report on statistical associations of all covariates used in an analysis, unless they are also a key focus of the study. in this case, the control variables do not appear to be critical to the study and therefore each one does not need to be interpreted in the text. If the authors do think interpreting these variables is necessary, then the frame and purpose should be modified to incorporate these.

9. In my prior comments, I mentioned that gender as a term was used to report sex (female, male). The authors reported they addressed this, but have not. Gender is reported in the abstract and referred to throughout the discussion and results sections.

10. In the manuscript info section, the authors reported that "data will be made available upon request" however in another area state the data are freely available. Please clarify

Reviewer #4: Having seen the manuscript's modifications, I agree with its publication, as it has been improved.

**Do you want your identity to be public for this peer review?** For information about this choice, including consent withdrawal, please see our Privacy Policy

Reviewer #1: No

Reviewer #2: No

Reviewer #3: No

Reviewer #4: **Yes: ** M. João Lima

---

## [Author Response · Author response to Decision Letter 2]

31 Aug 2025

31 August 2025

António Raposo

Academic Editor

PLOS ONE

We are pleased to have the reviewers’ comments on our manuscript entitled: “From Displacement to Hunger: How Migration Due to Conflict Affects Food Security in Yemen”. The comments were addressed, and the manuscript has been revised accordingly. The modifications (new edition) are highlighted in red in the revised manuscript.

Comments from the Reviewers: Modifications and explanations in response to the Reviewer’s comments are listed below:

Reviewer #1:

Critical Revisions Required:

DATA AVAILABILITY (MUST ADDRESS): Revert to the original public repository link (https://ghdx.healthdata.org/record/yemen-multiple-indicator-cluster-survey-2022-2023). And "Data available on request" violates PLOS policy.

Authors Response# We have amended the manuscript based on the guidelines of reviewer. [Page 22, Line 370-373]

Minor Suggestions:

Subgroup Analyses:

Briefly highlight key age/cohort findings (e.g., child migrants’ vulnerability) in the main text.

Authors Response# Age cohort tables are included in the paper. [See table 4, Page 13-14, Line 205-209]

Urban Context:

Strengthen policy linkages (e.g., connect "urban job creation" to Yemen’s National Food Security Strategy).

Authors Response# We have added some sentences to strengthen the policy linkage [Page 21, Line 342-348]

Policy Nuance:

Specify how "gender-sensitive interventions" (p. 345) align with Yemen’s cultural frameworks.

Authors Response# Thank you for this meticulous observation, we have added more detailed explanation in our revised manuscript. [Page 21, Line 359-361]

Reviewer #2:

1. The objectives presented in the introduction should be in past tense

Authors Response# We have amended the objectives in past tense. [Page 6, Line 89-96]

2. Yemen’s Central Statistical Organization led this comprehensive survey with technical assistance from UNICEF. “Were the authors part of these organizations?”

Authors Response# No, we are not part of these organizations.

3. Line 133….is the 1st variable the sum of 4th and 5th one?

Authors Response# Yes.

4. Line 183…. Sentence needs to be rephrased.

Authors Response# We have rephrased the sentence.

5. Line 187 represents or presents.

Authors Response# it should be “present”. We have corrected the term in the manuscript. [Page 11, Line 188]

6. Line 195 reported experience of hunger

Authors Response# We have omitted that line.

7. 202, 204, 211 sentences need revision to make those clear.

Authors Response# We have revised the lines based on the reviewer’s suggestion.

8. 239 Either this or thus is enough

Authors Response# We have revised the lines based on the reviewer’s suggestion.

Reviewer #3:

The authors present a revised version of the manuscript, which is in some instances improved. However, the authors did not address some of my most substantive feedback. My detailed comments are below.

1. The largest concern that I mentioned in my previous review that was not adequately addressed in this revision is regarding the use of causal claims and statements from cross-sectional data. The authors indicated in their response to reviews that they addressed this point by removing causal language throughout. However, this is not the case and causal language is used to discuss the findings throughout. I have pointed out specific instances of such language, although the authors should keep in mind that this is not an exhaustive list:

Lines 159-161 in statistical analysis: describes use of "two methods to investigate the impact of conflict migration on incidence of food insecurity." Using the term impact indicates causal and time sequential analyses; this is not as it is a cross-sectional analysis. The statement follows that "descriptive statistical analysis and .... negative binomial regression" are the methods employed, neither of which can support causal conclusions in this capacity.

Lines 207-208 results: "Migrating from another governorate did not have a statistically significant impact on..."

Lines 214-215: "Migrating due to conflict from another governorate had a statistically significant impact on..."

Lines 232-233 discussion: "had a considerable impact on both the prevalence of hunger and the frequency of going without food for an entire day."

Lines 299-300: "The research has highlighted that displacements caused by war and conflict are the primary cause of the increased food insecurity in Yemen."

Authors Response# All wordings have been changed.

2. There are also multiple instances in which the writing would benefit from clarity and careful consideration of framing. For example, the entire first paragraph of the discussion (Lines 229-237) is written in a manner that it is assumed that displacement causes food insecurity; however the authors also present this as the premise they were testing in the study. If it is already a foregone conclusion, it seems like doing the study would be unnecessary. In this case, better specification of the general premise that conflict-based displacement can worsen food insecurity, but this association has not been examined among people in Yemen specifically, would better facilitate the discussion intro and framing.

Authors Response# We have rearranged the discussion intro and framing following the reviewer’s suggestion. [Page 16, Line 241-246]

3. In the results, some of what is presented in the tables is repeated in the text; please summarize in the text and leave details for tables.

Authors Response# We have updated results.

4. Some statements in the results and throughout are awkward and difficult to understand. one example is lines 209-211: "If the individual migrated due to conflict and from another country, with statistical

robustness, it increases the risk of incidence of hunger (IRR: 2.100, 95% CI: 1.279–3.450) by

approximately more than two-folds...." what is statistical robustness in this context and why not instead just report the p value or CI?

5. Some of the methods and results is repetitive and should be streamlined. For example, Methodology section Lines 98-100 report the study purpose stated immediately above in the intro section.

Authors Response# We have made necessary update in results section.

6. Some of the language in the methods is imprecise and should be edited to improve clarity and specificity. For example, Line 143 "Additional variables are included to control mediating and moderating variables." There were no mediating or moderating analyses conducted. These variables were only included as covariates and therefore they should be labeled as such.

Authors Response# We have made necessary update in the manuscript.

7. Throught the manuscript, the tense switches from reporting in present tense to past tense and back again. Reporting of methods and results, which were conducted in the past, should be reported in past tense.

Authors Response# We have made necessary update in methods and results section.

8. In general, it is not necessary to report on statistical associations of all covariates used in an analysis, unless they are also a key focus of the study. in this case, the control variables do not appear to be critical to the study and therefore each one does not need to be interpreted in the text. If the authors do think interpreting these variables is necessary, then the frame and purpose should be modified to incorporate these.

Authors Response# We have made necessary update in the manuscript.

9. In my prior comments, I mentioned that gender as a term was used to report sex (female, male). The authors reported they addressed this, but have not. Gender is reported in the abstract and referred to throughout the discussion and results sections.

Authors Response# We have made necessary update in discussion and results section.

10. In the manuscript info section, the authors reported that "data will be made available upon request" however in another area state the data are freely available. Please clarify

Authors Response# We have amended the manuscript based on the guidelines of reviewer. [Page 22, Line-370-373]

Regards

Maruf Hasan Rumi

---

## [Decision Letter · Decision Letter 2]

18 Sep 2025

From Displacement to Hunger: How Migration Due to Conflict Affects Food Security in Yemen

PONE-D-25-26248R2

Dear Dr. Rumi,

We’re pleased to inform you that your manuscript has been judged scientifically suitable for publication and will be formally accepted for publication once it meets all outstanding technical requirements.

Kind regards,

António Raposo

Academic Editor

PLOS ONE

Additional Editor Comments (optional):

Reviewers' comments:

Reviewer's Responses to Questions

**Comments to the Author**

Reviewer #2: All comments have been addressed

2. Is the manuscript technically sound, and do the data support the conclusions?

Reviewer #2: Yes

3. Has the statistical analysis been performed appropriately and rigorously?

Reviewer #2: Yes

4. Have the authors made all data underlying the findings in their manuscript fully available?

Reviewer #2: Yes

5. Is the manuscript presented in an intelligible fashion and written in standard English?

Reviewer #2: Yes

Reviewer #2: The manuscript has been thoroughly evaluated. I find the work original, scientifically sound, and appropriate for publication in its current form. I have no concerns regarding research or publication ethics.

**Do you want your identity to be public for this peer review?** For information about this choice, including consent withdrawal, please see our Privacy Policy

Reviewer #2: No

---

## [Editor Report · Acceptance letter]

PONE-D-25-26248R2

PLOS ONE

Dear Dr. Rumi,

I'm pleased to inform you that your manuscript has been deemed suitable for publication in PLOS ONE. Congratulations! Your manuscript is now being handed over to our production team.

Kind regards,

on behalf of

Dr. António Raposo

Academic Editor

PLOS ONE